# MIRROR: Multi-model Inference and Regional Reasoning for Recognizing Annotation Errors

**Rohan Raju Dhanakshirur**[1] 🆔          ROHANRAJU.DHANAKSHIRUR@GEHEALTHCARE.COM
**Keerti K M** [1]                                    KEERTI.KM@GEHEALTHCARE.COM
**Prasad Sudhakar**[1]                          PRASAD.SUDHAKAR@GEHEALTHCARE.COM
**Chandan Aladahalli**[1]                 CHANDAN.ALADAHALLI@GEHEALTHCARE.COM
[1] *GE HealthCare, Bengaluru, India*

## Abstract

Accurate annotation is central to medical imaging AI. However, manual labeling remains error-prone due to operator variability, low contrast, subtle or transient anatomical boundaries, etc. These errors are often instance-dependent, arising precisely on the most clinically challenging frames, where conventional techniques such as confidence thresholding, repeated model training, etc. struggle to distinguish hard-but-correct samples from genuinely misannotated ones. Our analysis reveals that the modern regularized deep networks can tolerate random noise and they tend to exhibit consistent, convergent errors when the dataset label itself is incompatible with the underlying image content. Motivated by this, we introduce a simple architecture-agnostic detector that identifies potential misannotations by jointly requiring unanimous disagreement/model confusion across diverse models and high cross-model Grad-CAM agreement. Frames with low spatial consensus are instead attributed to heterogeneous model errors rather than label corruption. Across MRI, and X-ray datasets with 5% synthetic label corruption, this dual-consistency criterion recovers mislabeled samples with 93%, and 96% F1-score, outperforming the best performing state-of-the-art noisy-label baselines by 8.14% and 3.23% respectively. Qualitative examples further show that flagged cases exhibit stable saliency across models. These results suggest that cross-model semantic alignment against the provided label is a reliable and interpretable indicator of annotation error, enabling efficient, high-precision data auditing without requiring clean subsets or repeated retraining.

**Keywords:** Medical Imaging, Noisy Labels, Annotation Error Detection, Multi-Model Consistency, Explainable AI

## 1. Introduction

Manual annotation in medical imaging is challenging due to ambiguous anatomy, noise, and operator variability, resulting in frequent instance-dependent errors that mimic hard, but-correct samples. Traditional noisy-label signals such as confidence thresholding, loss, or uncertainty fail to separate these cases reliably. Existing approaches such as Confident Learning (Northcutt et al., 2021), ReCoV (Chen et al., 2024), uncertainty-based filtering (Xu et al., 2023; Shama et al., 2026), or training-dynamic methods (Kim et al., 2024)

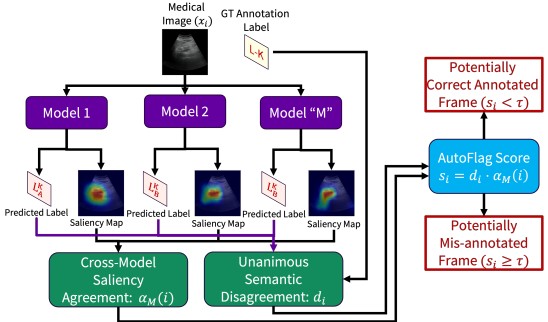

Figure 1: **MIRROR Overview.** Multi-model predictions and Grad-CAM maps jointly produce the AutoFlag score $s_i = d_i \alpha_M(i)$ for flagging likely misannotations.

Table 1: Detection performance under synthetic corruption for MRI (Cheng et al., 2015) and X-ray (Chowdhury et al., 2020) datasets.

|  | MRI | | X-ray | |
|---|---|---|---|---|
| **Method** | **Prec** | **Rec** | **Prec** | **Rec** |
| (Northcutt et al., 2021) | 0.29 | 0.86 | 0.77 | 0.87 |
| (Xu et al., 2023) | 0.86 | 0.86 | 0.93 | 0.93 |
| (Chen et al., 2024) | 0.45 | 0.90 | 0.59 | 0.97 |
| (Hoang et al., 2024) | 0.43 | 0.43 | 0.62 | 0.62 |
| (Shama et al., 2026) | 0.53 | 0.53 | 0.41 | 0.41 |
| **Proposed** | **0.93** | **0.93** | **0.95** | **0.97** |

require clean subsets, repeated retraining, or lack spatial interpretability, making them less suited for fine-grained medical imaging errors.

A widely observed phenomenon motivates our approach: for a given specific image where labels are incompatible with image content, diverse deep networks tend to make *consistent* errors and focus on the same anatomical region. We leverage this property to identify misannotations using two complementary criteria: unanimous semantic disagreement and cross-model alignment of Grad-CAM evidence. This results in a lightweight, interpretable, and dataset-agnostic signal for surfacing questionable annotations at scale.

## 2. Method

Let $\mathcal{X} = \{(x_i, y_i)\}_{i=1}^{N}$ denote a dataset with potentially corrupted labels. We train $M$ diverse classifiers $\{f_m\}$ using standard ERM. For each model, let $\hat{y}_i^{(m)}$ be its top-1 prediction with $c_i^m$ being its prediction confidence $\in [0, 1]$ and $S_i^{(m)}$ its Grad-CAM map.

**Unanimous Semantic Disagreement:** We first identify samples for which all models reject the annotated label, or when each of the models are confused on the samples (i.e., when prediction confidence is low): $d_i = \mathbb{1}\left[\{\hat{y}_i^{(1)} = \cdots = \hat{y}_i^{(M)} \neq y_i\} || \{c_i^m < \tau^*, \forall m\}\right]$.

**Cross-Model Saliency Agreement:** To quantify spatial consistency, we compute: $\alpha_M(i) = \frac{\sum_p \min_m S_i^{(m)}(p)}{\sum_p \max_m S_i^{(m)}(p) + \varepsilon}$, which is high when all models highlight the same anatomical region.

**AutoFlag Score:** Our final misannotation score is: $s_i = d_i \alpha_M(i)$, and samples with $s_i \geq \tau$ are flagged for audit. This dual requirement suppresses model-specific overfitting while surfacing anatomically coherent, label-incompatible frames.

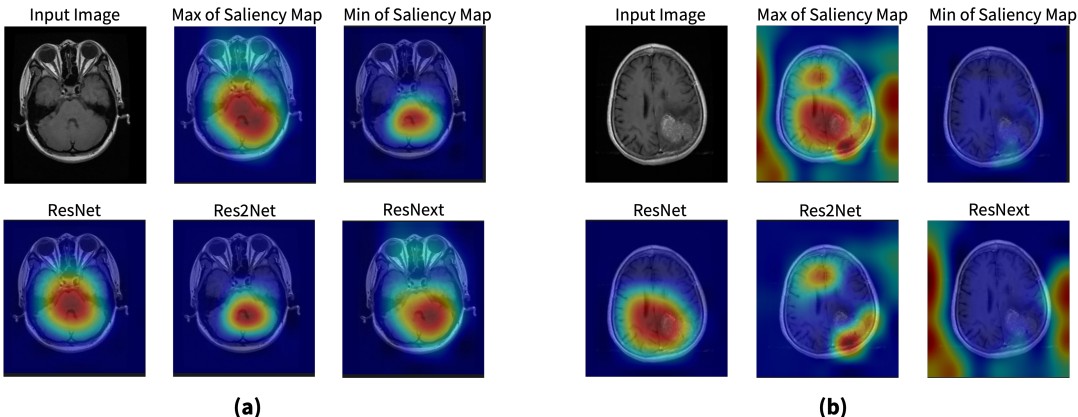

Figure 2: **Examples of cross-model semantic and spatial consistency.** (a) Potentially misannotated frame: models focus on the same region, yielding high $\alpha_M(i)$ and $s_i$. (b) Correctly annotates frame: spatial evidence varies across models, producing low $\alpha_M(i)$ and low $s_i$.

## 3. Results

Across MRI (Cheng et al., 2015) and X-ray (Chowdhury et al., 2020), datasets with 5% corruption, our method consistently outperforms SOTA detectors. MIRROR achieves recall values of 93.3% and 97.5% while maintaining high precision (Table 1). Unlike single-model heuristics that often misidentify difficult samples as noisy, MIRROR explicitly requires both semantic unanimity and anatomical alignment, yielding more stable behavior.

**Ablation Summary:** Although full ablations cannot be included due to space constraint, we summarize the main findings: (i) using $d_i$ alone has high recall but low precision; (ii) $\alpha_M(i)$ alone lacks discriminability; (iii) combining both yields substantial improvements. Performance saturates at $M = 3$, indicating that small, diverse ensembles are sufficient.

**Qualitative Analysis:** Misannotated samples show strong cross-model semantic and spatial consensus. Tough but correctly annotated samples show dispersed saliency, resulting in lower $\alpha_M(i)$ and lower $s_i$ (Figure 2).

## 4. Conclusion

We presented MIRROR, a simple and interpretable framework for detecting annotation errors by combining unanimous multi-model disagreement with cross-model Grad-CAM alignment. Experiments across modalities show improved recall and stable performance without requiring clean subsets or retraining. By surfacing images whose evidence patterns contradict provided labels, MIRROR provides a practical, anatomically grounded tool for efficient dataset auditing in clinical AI pipelines.

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
