# OpenReview forum: "MIRROR: Multi‑model Inference and Regional Reasoning for Recognizing Annotation Errors"
_MIDL.io/2026/Short_Papers — MIDL 2026 - Short Papers Poster_

### Official Review · Reviewer_ogYV · 2026-04-23
**Simple and well presented paper**

**Rating:** 4
**Confidence:** 3

**Review:**

- Quality/Clarity: The paper presents its main idea and results clearly, making the contribution easy to follow.

- Significance/Originality: The combination of model disagreement and saliency-based agreement for label noise detection is reasonable and shows competitive performance against selected methods. I could not judge how well the baselines chosen represent the literature on label noise detectors.

**Summary:**

The paper proposes combining model disagreement with saliency maps (e.g., GradCAM) to detect label noise. The core idea is that when predictions and saliency maps agree across models, the corresponding label is likely misannotated. They evaluated their method on two datasets and compared it against five methods from the literature, and showed noticeable improvements in Precision and Recall.

**Strengths:**

- The main idea and experimental results are communicated clearly and concisely.
- The method's evaluation was somewhat extensive. They evaluated their method on two datasets and compared it against five methods from the literature. It was also ablated, but the results were not shown due to limited space.
- The method, despite its simplicity, compares favorably against methods from the literature.

**Weaknesses:**

- The paper does not describe how the threshold for classifying a sample as noisy or clean is determined. This is an important methodological detail that should be addressed.
- The method is likely not robust in settings where models are confidently wrong, for example, when spurious features are strongly correlated with the label, or under adversarial conditions. This limitation is not discussed.

**Justification Of Rating:**

The paper presents a clear and well-motivated idea with promising results. Addressing the threshold selection and discussing limitations in robustness would considerably strengthen the contribution.

---

### Decision · Program_Chairs · 2026-05-08

Accept (Poster)